# Interventions to reduce inequalities in avoidable hospital admissions: explanatory framework and systematic review protocol

Sarah Sowden [ID],[1] Behrouz Nezafat-Maldonado,[1] Josephine Wildman,[1] Richard Cookson,[2] Richard Thomson,[1] Mark Lambert,[3] Fiona Beyer,[1] Clare Bambra[1]

[1]Population Health Sciences Institute, Newcastle University, Newcastle upon Tyne, UK
[2]Centre for Health Economics, University of York, York, UK
[3]North East Centre, Public Health England, Newcastle upon Tyne, UK

**Correspondence to**
Dr Sarah Sowden;
sarah.sowden@newcastle.ac.uk

## ABSTRACT

**Introduction** Internationally there is pressure to contain costs due to rising numbers of hospital admissions. Alongside age, socioeconomic disadvantage is the strongest risk factor for avoidable hospital admission. This equity-focussed systematic review is required for policymakers to understand what has been shown to work to reduce inequalities in hospital admissions, what does not work and where the current gaps in the evidence-base are.

**Methods and analysis** An initial framework shows how interventions are hypothesised to reduce socioeconomic inequalities in avoidable hospital admissions. Studies will be included if the intervention focusses exclusively on socioeconomically disadvantaged populations or if the study reports differential effects by socioeconomic status (education, income, occupation, social class, deprivation, poverty or an area-based proxy for deprivation derived from place of residence) with respect to hospital admission or readmission (overall or condition-specific for those classified as ambulatory care sensitive). Studies involving individuals of any age, undertaken in OECD (Organisation for Economic Co-operation and Development) countries, published from 2000 to 29th February 2020 in any language will be included. Electronic searches will include MEDLINE, Embase, CINAHL, Cochrane CENTRAL and the Web of Knowledge platform. Electronic searches will be supplemented with full citation searches of included studies, website searches and retrieval of relevant unpublished information. Study inclusion, data extraction and quality appraisal will be conducted by two reviewers. Narrative synthesis will be conducted and also meta-analysis where possible. The main analysis will examine the effectiveness of interventions at reducing socioeconomic inequalities in hospital admissions. Interventions will be characterised by their domain of action and approach to addressing inequalities. For included studies, contextual information on where, for whom and how these interventions are organised, implemented and delivered will be examined where possible.

**Ethics and dissemination** Ethical approval was not required for this protocol. The research will be disseminated via peer-reviewed publication, conferences and an open-access policy-orientated paper.

## Strengths and limitations of this study

► The study employs a rigorous international gold-standard methodology (Preferred Reporting Items for Systematic Reviews and Meta-Analyses - Equity (PRISMA-E)) to undertake an equity-focussed systematic review and therefore will provide a comprehensive overview of existing literature on interventions that reduce socioeconomic inequalities in avoidable hospital admissions.

► By following PRISMA-E guidelines this study includes a framework to explain how interventions might reduce inequalities in hospital admissions which will be useful to decision-makers.

► The review adopts a newly developed and validated comprehensive search strategy for identifying studies focussed on equity issues and is therefore at the forefront of developments in equity-based systematic reviewing practice.

► One limitation is that we will not conduct full-text hand-searching and so may miss studies which have not made any reference to equity-related terms in either the title or abstract.

► Another limitation is that studies may be too heterogeneous to obtain combined effect estimates through meta-analysis.

**PROSPERO registration number** CRD42019153666.

## INTRODUCTION

Due to pervasive socioeconomic inequalities in health,[1] people in disadvantaged neighbourhoods are more in need of care. Often, however, they have worse access to, experience of and outcomes from healthcare—even in universal health systems.[2] Potentially avoidable hospital admission for chronic conditions are an international concern and, for example, account for over 37 million bed days each year across the European Union.[3] Social and economic inequality is a likely contributor to such admissions—in all countries

regardless of the health system financing scheme, population characteristics or coverage rates. Alongside age, socioeconomic disadvantage is known to be the strongest risk factor for avoidable hospital admission.[4] To take England as an example, in 2015, 263 894 excess avoidable emergency admissions were associated with socioeconomic inequality[5] and excess hospitalisations associated with socioeconomic inequality cost the National Health Service £4.8 billion per year.[6] Reducing pressures on hospital services therefore requires addressing the socioeconomic differences in health risks and behaviours that give rise to inequalities in avoidable hospital admissions.[7]

Different ethnic groups experience different avoidable admission rates, and the interrelationship between ethnicity and socioeconomic status is complex and context specific.[8] There are national variations in rates of avoidable admissions[3] as well as local variation in avoidable admission socioeconomic inequality[9 10] in Organisation for Economic Co-operation and Development (OECD) countries.

## Policy context

There is growing pressure to contain costs and prevent disruptions in elective hospital care due to a rising number of hospital admissions across OECD countries.[11] Getting health and social care systems working better together to reduce avoidable admissions[7 12 13] is a common, key priority in diverse health systems.[14 15] Addressing inequalities and improving the integration of primary and secondary services has a high policy profile in the UK[16] and internationally.[17] For example, WHO Europe Health 2020 framework aims to improve health for all and reduce health inequalities through investment in health, tackling major disease burdens, strengthening people-centred health systems and public health capacity and creating supportive environments and resilient communities.[18]

Some countries have explicit statutory duties to address health and healthcare inequalities. In Spain, the Royal Decree Law 7/2018,[19] recognises access to the National Health System as a fundamental right of every person in Spain. In addition, Spanish health regulations such as The General Health Act (1986)[20] and the National Health System Cohesion and Quality Act (2003)[21] aim to overcome health inequalities and guarantee equality of access to public healthcare services. Similarly, the Health and Social Care Act 2012[22] in England mandates each local area in England to take steps to address inequalities in healthcare outcomes and tackling health inequalities is seen as critical in ensuring the long-term sustainability of the UK national health service.[23]

There are promising interventions to reduce avoidable hospital admissions in areas including education, self-management, rehabilitation and telemedicine.[13] However, existing systematic reviews have examined *only* the effects of interventions on reducing levels of hospital admissions *overall*, as opposed to the effects on *inequalities* in avoidable hospital admissions. The differential effectiveness of interventions across socioeconomic groups in unknown, including a lack of evidence on any interventions aimed at reducing inequalities or targeted in their delivery to disadvantaged groups.[4 13]

It is critical for policy-making in this area that evidence of the effectiveness of different types of interventions at tackling inequalities is systematically reviewed. There is currently a lack of accessible policy and practice ready evidence on what works in terms of interventions to reduce inequalities in avoidable hospital admissions. Conversely, there is limited understanding of what interventions should be avoided; frequently well-intentioned interventions may increase rather than decrease inequalities.[24 25] The organisation and implementation of such interventions is also important to understand. Little is known about the effectiveness of system wide approaches to reducing avoidable admissions[13] or the impact of wider contextual factors outside the direct control of local health and care provision; real world interventions are rarely implemented in isolation and the complex interaction of interventions with the particular context in which they are embedded determines outcomes.[12 26 27] Internationally, there is a lack of research on potential explanations for the apparent variation in healthcare equality performance between local areas required for those working in policy and practice to learn quality improvement lessons.[28]

Against this backdrop, this systematic review will address this deficit in the knowledge base by reviewing primary studies of the effectiveness of interventions in reducing socioeconomic inequalities in avoidable admissions. It will consider interventions (across population health and policy-level, community, service-based and integrative domains of action) which might reduce socioeconomic inequalities in avoidable admissions, highlight any gaps in the evidence base and, for any interventions identified, seek to establish how such interventions are organised, implemented and delivered.

## Intervention framework

In line with PRISMA-E[29–31] guidelines, as the first stage of this equity-focussed review we have developed a framework[29–34] for how inequalities in avoidable hospital admissions might be tackled (figure 1). The framework outlines the pathways through which interventions are hypothesised to operate to result in reducing socioeconomic inequalities in avoidable hospital admissions over time. This framework has adopted a system-level focus to capture the complexity of place-based working.[35–37] It has been developed from existing frameworks for avoidable admissions[5] and for addressing health and healthcare inequalities.[38–42] The intention is for this to be further revised iteratively[34] as evidence from the systematic review emerges and in consultation with patient, public and policy and practice partners. In this systematic review the interventions will be grouped according to this framework (with recognition that some interventions may be cross-cutting). The term intervention

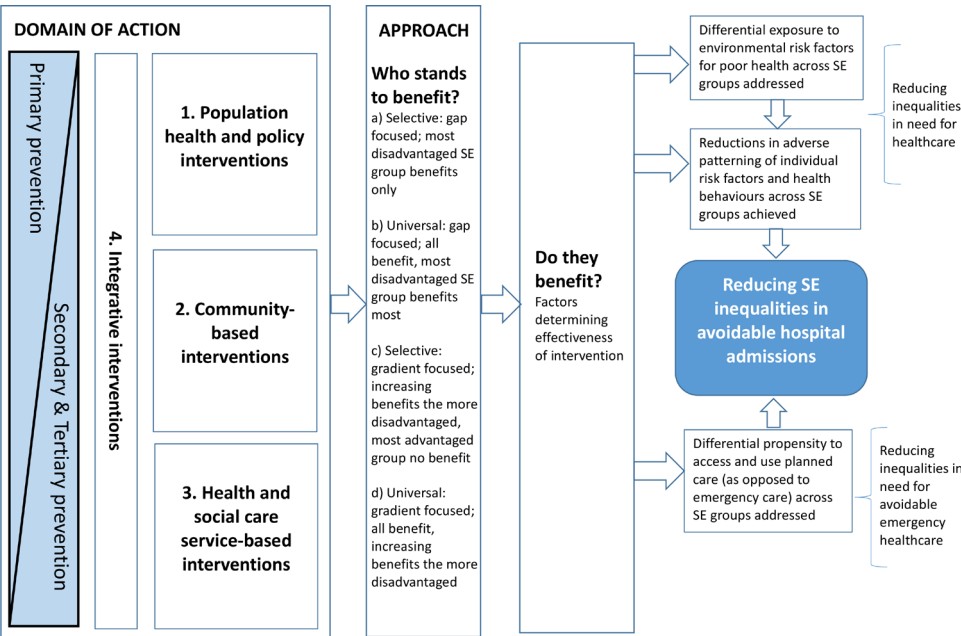

**Figure 1** Framework for addressing socioeconomic (SE) inequalities in avoidable hospital admissions.

is used to refer to any programme, policy, strategy, initiative, scheme or activity that has been implemented where either: (a) the outcome of the action taken has been measured in terms of avoidable hospital admission across socioeconomic groups or (b) the intervention was focussed specifically on socioeconomically disadvantaged areas or groups and where the outcome of the intervention was measured in terms of avoidable hospital admissions in this group.

Tackling socioeconomic inequality in healthcare is challenging,[43 44] requiring action on a broad scale[1 2 39 42 45] and health and social care systems working together with other agencies[28 38 39 46] to bring about change. There is increasing recognition among policymakers and commissioners that to tackle complex issues such as avoidable admissions effectively and to reduce inequalities in health and healthcare requires integrated policy, strategy and interventions across different domains of action (population health and policy-level, community-based, health and social care service-based as well as those focussed on integrating activity).[2 39] Based on this understanding, this framework has characterised interventions by their domain of action (1 to 4) and their approach to tackling inequalities (a to d). There are four broad domains of action[16] and the orientation of activity to primary, secondary and tertiary prevention[47] shifts accordingly across the domains of action (figure 1):

1. Population health and policy-level interventions – legal, fiscal, structural, organisational, environmental and policy interventions that seek to change health-related behaviours or to modify the social and economic determinants of health.[48] This includes interventions with other goals that bring about such changes as a by-product.[48]

2. Community-based interventions – community resources and assets, for example, local area coordination initiatives.

3. Health and social care service-based interventions – in this context these might include interventions considered in previous reviews of avoidable admissions[13] (for example, disease and medication management and education programmes, vaccination, structured discharge planning, comprehensive geriatric assessment, virtual wards, hospital at home initiatives) as well as primary care (provision, access, quality, continuity), benchmarking, predictive risk-modelling and case-management interventions.

4. Integrative interventions – interventions that create greater synergy and closer working between domains 1 to 3 or within 3 (between, for example, primary and secondary healthcare and social care). For example, social prescribing initiatives facilitate closer working between domains 2 and 3.

According to Benach *et al*[40] interventions to address inequalities may adopt one of four different approaches:

a. Targeted intervention to those in most disadvantaged socioeconomic group only, focussed on closing the health gap (selective: gap focussed).

b. Universal intervention, with additional focus on those in the most disadvantaged socioeconomic group to close the health gap (universal: gap focussed).

c. Redistributive intervention, a universally applied intervention, but where the extent of benefit increases across the social gradient, such that the most socioeconomically advantaged are not expected to benefit at all from the intervention due to the lack of need for it (selective: gradient focussed).

d. Proportionate universalism, universal intervention benefitting all, but with increasing benefits of intervention across the social gradient (universal: gradient focussed).

Here, we provide three hypothetical illustrative examples of the application of this framework in the context of reducing socioeconomic inequalities in avoidable hospital admissions. First, there may be a social prescribing intervention (domains 2 and 4) which is facilitated through citizens advice services and serves those who have a limiting long-term condition and are in receipt of unemployment or other welfare benefits (approach c). This intervention is designed, financed and implemented in such a way that it is successful in reducing avoidable hospital admissions for these individuals. Second, there may be a water fluoridation intervention (domain 1). This intervention benefits the whole population in terms of reductions in hospitalisations resulting from tooth decay but with increasing benefits across the social gradient as need for the intervention is greatest in lower socioeconomic groups where tooth health is poorest (approach d). Frequently well-intentioned interventions may increase rather than decrease inequalities.[24 25] So in a final example, there may be a virtual ward intervention (domain 3) introduced for patients in a local area that fails to adopt any approach to addressing inequalities (a to d), and so while the intervention may lead to a reduction in avoidable hospital admissions per se, it may maintain or increase the disparity in avoidable admissions between socioeconomic groups within this locality. The application of the framework to these three hypothetical example interventions would therefore be summarised as follows; social prescribing intervention (2c, 4c), water fluoridation (1d), virtual hospital ward intervention (3x), where x represents no approach to addressing inequalities has been adopted for the intervention.

## METHODS AND ANALYSIS

The review will be carried out following established criteria for the good conduct and reporting of equity-focussed systematic reviews using PRISMA-E guidelines[29–31] and reporting here conforms to the standards of the Preferred Reporting Items for Systematic Reviews and Meta-analysis Protocols (PRISMA-P, see online supplementary file 1).

Preliminary searches were carried out in September and October 2019 and the study registered with the PROSPERO International Prospective Register of Systematic Reviews[49] on 21st October 2019. The review is planned to commence in March 2020 and is anticipated to take 1 year to complete.

## Patient and public involvement

An advisory panel comprising collaborators from the English public health and healthcare system, national and international research communities, social and voluntary sector and patient and public representatives will guide the research. The early engagement of patients and members of the public through the National Institute for Health Research (NIHR) North East Research Design Service consumer panel at the stage of requesting NIHR funding to carry out this research has been instrumental in shaping this systematic review protocol. Patient and public involvement (PPI) members helped to strengthen the articulation of the study rationale. We received a powerful mandate from panel members to undertake the research based on their unanimous belief about its critical importance to the healthcare system and wider society. It became apparent during our discussions that the legal commitment to reducing inequality in service provision and outcome in England was not widespread public knowledge and panel members suggested this was given more prominence in the background rationale.

Through the collaborator advisory panel, patients and the public will contribute, alongside academics, policy and practice partners, to the iterative further development of the framework (figure 1). In order to ensure meaningful participation of PPI members as the systematic review develops we will meet regularly highlighting areas for input. Outputs from this systematic review, including a workshop event will be developed in close collaboration with both PPI and policy and practice partners.

## Systematic review questions

► What interventions (population health, community-based, service-based and integrative) reduce socioeconomic inequalities in avoidable hospital admissions?
► What interventions (population health, community-based, service-based and integrative) maintain or increase socioeconomic inequalities in avoidable hospital admissions?

## Objectives

This study has two objectives:
1. To systematically review the effectiveness of interventions (population health, community-based, service-based and integrative) in reducing socioeconomic inequalities in avoidable hospital admissions.
2. For interventions identified through the systematic review, establish where possible, how these specific interventions are organised, implemented and delivered.

## Eligibility criteria

Studies will be selected according to the criteria outlined in table 1.

## Interventions

The review will examine interventions across the range of domains of action (population health and policy-level, community-based, service-based and integrative) which might reduce socioeconomic inequalities in avoidable hospital admissions in individuals of any age. It will also uncover those interventions which may serve to maintain or increase inequalities.

The review will use the intervention framework and group interventions by their domain of action and by

**Table 1** Study inclusion and exclusion criteria

|  | Inclusion | Exclusion |
|---|---|---|
| Participants (P) | All persons of any age in OECD countries | Non-OECD countries |
| Intervention (I) | Any intervention (programme, policy, strategy, initiative, scheme or activity) across the range of domains of action (population health and policy-level, community-based, service-based and integrative) |  |
| Comparison (C) | No intervention; usual care or practice; other interventions |  |
| Outcomes (O) | For universal interventions; <br>▶ Numbers, rates, ratios, standardise mean differences or a gradient measure <br>▶ Of hospitalisation overall or condition-specific hospital admission/readmission for any ambulatory care sensitive conditions <br>▶ Across socioeconomic groupings <br>For interventions targeted exclusively at socioeconomically disadvantaged populations; <br>▶ Hospitalisation overall or condition-specific hospital admission/readmission for any ambulatory care sensitive condition. | Studies that do not include a subanalysis of effectiveness by socioeconomic group or a measure of inequality such as an absolute gradient of inequality (N/A to interventions targeted exclusively at socioeconomically disadvantaged populations) |
| Study type | ▶ Randomised controlled trials <br>▶ Non-randomised or uncontrolled trials <br>▶ Prospective and retrospective cohort studies (with and/or without control groups) <br>▶ Time-series data studies <br>▶ Panel data studies (with and/or with control groups) <br>▶ Case-control studies <br>▶ Other ecological studies | ▶ Descriptive studies reporting solely on avoidable admission prevalence (by person, place and time) <br>▶ Qualitative studies <br>▶ Editorial <br>▶ Commentary <br>▶ Expert opinion |
| Study period | Published in the last 20 years (2000–2020) | Literature published before 2000 |
| Study reporting language | Any language |  |

N/A, not applicable; OECD, Organisation for Economic Co-operation and Development.

their approach to tackling inequalities (figure 1). Where possible the interventions will be grouped according to these criteria with the acknowledgement that some interventions might be cross-cutting.

## Outcomes

'Avoidable admissions' are those for which timely and effective ambulatory care can prevent the need for hospitalisation.[50] Different studies use different lists of ambulatory care sensitive conditions, reflecting different types of illness (for example, whether illness is considered chronic or acute), different types of care (for example, primary care, secondary care, community care) and differences in clinical coding definitions.[3 50–53] Avoidable admissions are sometimes referred to as preventable admissions. Our definition of outcome is general and permissive of all this definitional variation within and between countries. For targeted interventions focussed exclusively on socioeconomically disadvantaged populations, studies reporting a global measure of hospitalisation and those reporting condition-specific[54] hospital admission measures for any ambulatory care sensitive condition will be included. For interventions not specifically targeted at socioeconomically disadvantaged populations, studies will be included only if the differential effect of the intervention by socioeconomic status (education, income, occupation, social

class, deprivation, poverty or an area-based proxy for deprivation derived from place of residence) with respect to hospitalisations is reported. For example, using a comparison of numbers, rates, ratios, standardise mean differences or a gradient measure of hospitalisation overall or condition-specific hospital admission/readmission for any chronic ambulatory care sensitive condition across socioeconomic groupings.

## Study exclusion

The search will be limited to studies published in the last 20 years (since 2000) that have been carried out in an OECD country to ensure contemporaneity and a degree of commonality in health system maturity and socioeconomic and demographic context. We recognise that OECD countries will have different characteristics including different health systems and socioeconomic situations and we will interpret the interventions that we find taking into account this variation in context.

## Study design

A rigorous and inclusive international literature search for experimental, quasi-experimental and analytical observational studies will be conducted. This will include randomised and non-randomised controlled trials, non-controlled trials, prospective and retrospective cohort

studies (with and/or without control groups), time series data studies, panel studies (with and/or with control groups), case-control studies and other ecological studies (for example studies using administrative data) of the effectiveness of interventions at reducing inequalities in avoidable admissions to hospital. In the case of non-randomised controlled trial studies the authors will make a judgement as to whether any association reported may be considered causal, taking into account the type of study undertaken and following a thorough assessment of study quality and risk of bias (see risk of bias section below).

### Information sources and search strategy

Published studies in any language meeting the inclusion criteria will be sought by searching peer-reviewed literature in electronic databases including (host sites given in parentheses): MEDLINE (Ovid), Embase (Ovid), CINAHL (EBSCO), Cochrane CENTRAL (Wiley), Science Citation Index (Web of Knowledge), Social Science Citation Index (Web of Knowledge), Conference Proceedings Citation Index – Science (Web of Knowledge), Conference Proceedings Citation Index – Social Science (Web of Knowledge). All searches will be re-run prior to final analyses and any further studies identified retrieved for inclusion. Please see online supplementary file 2 for the preliminary MEDLINE search strategy.

Web of Knowledge Citation Indices will be used to undertake full citation searches of included studies. Electronic database searches will be supplemented with website searches, included but not restricted to the Kings Fund and the Nuffield Trust. We will request relevant information on unpublished and in-progress research from key experts in the field, identified through the collaborator network for this systematic review.

### Study selection

All articles identified through the search will be uploaded into EPPI-Reviewer 4 software.[55] Duplicates will be removed automatically and manually. Three reviewers (SS, BN-M and JW) will screen titles and abstracts (if available) of the retrieved articles to assess eligibility for inclusion. The initial 10% of the papers will be double-screened independently and then reviewed to ensure a consistent approach to applying the inclusion criteria is being adopted. After the initial selection based on title and abstract screen full texts of potentially eligible articles will be obtained and screened independently by two reviewers to assess eligibility for final inclusion. Disagreements at any stage will be resolved through consultation with a fourth reviewer (FB).

### Data extraction

Data for included studies will be extracted by two reviewers using a bespoke form within EPPI-Reviewer4 software.[55] The data extraction of one reviewer will be checked by the other and visa versa. Any discrepancies will be resolved through discussion between the reviewers

and, if consensus is not reached, with the wider project team (FB, RT, RC, ML, CB). The data extraction form will be piloted using the first five eligible articles and modified if required (see online supplementary file 3). The data extracted from each article will include, but not restricted to, author, publication year, funding source, study design, study setting including characteristics of the health system of the country, study period, number of included participants/areas, study population characteristics (including for example age and ethnicity), intervention domain (1 to 4), intervention approach (a to d), intervention description, definition and measurement of socioeconomic status, outcome measure(s), covariates used for adjustment (certain study types), reported limitations and key conclusions. It is anticipated that studies will characterise socioeconomic groupings differently and measure equity in a variety of ways. Therefore, the data collection tool will be designed to ensure all ways of measuring socioeconomic status and differential outcomes across groups will be recorded. If relevant information cannot be retrieved from the published articles, where possible, the manuscript authors will be contacted to request additional data.

For included studies, data on the organisation, implementation and delivery of these interventions will be sought from the manuscript and any associated papers through a linked citation search. In conjunction with an OECD healthcare system typology,[56] we will use data collected from the OECD Health System Characteristic survey[57] to explore potential differences between countries and we will classify them based on the most common health coverage that is implemented, for example, autonomic coverage, compulsory coverage, voluntary coverage or no coverage. The Template for Intervention Description and Replication - Population Health and Policy (TiDieR-PHP)[58] or TIDieR-PHP[48] checklist will be used to extract relevant contextual information. Any associated process evaluations (quantitative and qualitative) exploring how and why such interventions work will be reviewed. Examples of the implementation components that will be examined include theoretical underpinning, implementation context, experience level of the intervention team (planners and implementers), consultation and/or collaboration processes (planning and delivery stages) and resources (for example time, money, staff and equipment). The systematic review evidence combined with the implementation analysis will provide insights into what works to reduce socioeconomic inequalities in avoidable admissions to hospital, why and how.

### Risk of bias assessment

The methodological quality of the included studies will be appraised independently by two reviewers using the Effective Public Health Practice Project (EPHPP) tool,[59] chosen because the same tool can be used across a range of quantitative study designs. Any disagreements in reviewer quality assessment judgements will be resolved through discussion between the reviewers, and if necessary with

a fourth reviewer (FB). Using the EPHPP tool, the two reviewers will make their own scientific value judgements about whether quasi-experimental methodology studies are sufficiently well designed to draw robust causal inferences, referring controversial cases to the wider review team. In rare cases, for example, robust causal inferences can be derived from uncontrolled before-after studies (and indeed from cross-sectional correlations). The quality appraisal criteria will be used for descriptive purposes, to highlight variations between studies and to contribute to the assessment of overall strength of evidence in this topic area.

### Analyses and synthesis

Data will be presented narratively as well as in tables (where possible) to describe the study designs, population and findings and to address each of the research questions. The main analysis will examine the effects of interventions across the domains of action on socioeconomic inequalities in avoidable hospital admission, categorising the approach adopted to address inequalities for each intervention, using the multidimensional framework outlined in figure 1. Absolute and relative differences in outcome between socioeconomic groups, at baseline and follow-up, or across place and time (depending on study design), will be considered. Data will be summarised statistically using meta-analysis of effect measures across socioeconomic subgroups, where appropriate, and if studies are deemed sufficiently homogeneous to combine. Similarly, for studies exclusively focussed (in terms of study setting) on socioeconomically disadvantaged population groups, the feasibility of conducting meta-analysis for studies within this category will be assessed. Where there are sufficient numbers of studies, funnel plots and, where applicable, appropriate statistical tests will be used to assess publication bias and small study effects. EPPI-Reviewer 4[55] will be used to assist in the analysis.

Where applicable and feasible, we will synthesise results for different population subgroups separately (for example, in regards to children compared with the elderly, acute compared with chronic ambulatory care sensitive conditions) where the differential impact of interventions across socioeconomic groups on hospitalisations was considered.[54] Where data permit, we will conduct further demographic subgroup analysis of the socioeconomic patterning of effect by age, gender and ethnicity. We will report our analyses in accordance with the PRISMA-E guidelines.[29–31]

### DISCUSSION

The review will consider interventions which reduce socioeconomic inequalities in avoidable admission to hospital. The review will also serve as a mapping exercise of the types of interventions that have been evaluated in relation to tackling inequalities in avoidable admissions, thereby highlighting any gaps in the evidence base.

For included studies, further scrutiny of any published information about them, will seek to uncover how these interventions are organised, implemented and delivered. Context is increasingly recognised as important; real-world interventions are rarely implemented in isolation and the complex interaction of interventions with the particular social context in which they are embedded determines outcomes.[12 26] However, the assessment of context and implementation has not featured strongly in previous reviews.[13] It is essential for policy and practice partners to learn not only that an intervention works to reduce inequalities in avoidable hospital per se, but why and how it managed to do this. This will enable local decision makers to compare the contextual situation to their own to make an assessment of both feasibility and likelihood of success of adopting a similar intervention in their specific socio-political, economic, cultural landscape. The ability of the review to provide this information to assist in implementation will be constrained by the extent to which this contextual information is examined and reported on in the primary studies. For example, an intervention to address socioeconomic inequalities may work for a specific ethnic group (or other subpopulation) but will not work in another, and if a subanalysis by ethnicity of the differential effectiveness of the intervention across socioeconomic groups is not carried out, or an insufficient description of the targeted population who were exposed to the intervention is not provided, this nuance will be lost.

The study design inclusion criteria in the review are broad, given that while trials of service-based (domain 3), and even community-based (domain 2) interventions may be likely, we expect a lack of experimental studies in relation to population health and policy-level (domain 1) and many integrative interventions (domain 4) given these tend not to be easily or routinely evaluated using experimental study designs.[41 60]

Only a minority of intervention studies published report any outcome by a social determinant of health and locating these studies has previously been hampered by the absence of validated equity search filters.[61] Reflecting this, the PRISMA-E guidance recommended that systematic reviewers avoid filtering searches on equity terms due to poor indexing which would lead to relevant studies being missed and advised hand-searching on these terms instead.[30] However, in 2018 an equity-focussed search strategy was built and assessed for sensitivity against a gold-standard set of equity focussed studies.[61] Our review adopts this newly developed and validated comprehensive search strategy for identifying studies focussed on equity issues and is therefore at the forefront of developments in equity-based systematic reviewing practice. Nevertheless, a limitation remains that by not conducting full-text hand-searching on equity-related terms the review may miss studies which have not made any reference to this aspect of their analysis in either the title or abstract.

Our extensive search strategy, combined with the inclusive study design criteria, will ensure that a sizeable

literature will be located for synthesis. A 2012 series of reviews of interventions to reduce unplanned hospital admissions found 1530 controlled trials.[13] While we recognise that the literature on the effects of interventions on socioeconomic inequalities in avoidable hospital admissions is likely to be smaller, we will maximise the likelihood of locating relevant studies by taking a more inclusive approach to study design and evaluate interventions targeted at socioeconomically disadvantaged groups or areas as well as studies that include comparative data on the differential effects of an intervention across socioeconomic groups. The size of the available evidence base will also be extended, because we will adopt a place-based, whole systems perspective, considering several domains of action (population health and policy-level, community-based, service-based and integrative).

## Ethics and dissemination

Ethical approval is not required for this protocol or for the systematic review because it does not involve primary data collection.

We anticipate the findings for this review will contribute to an improved understanding of interventions which reduce socioeconomic inequalities in avoidable hospital admissions. Once the evidence has been synthesised a workshop will be held with invited health and social care commissioners and providers, the public health practice community who hold responsibilities for reducing health and healthcare inequalities; relevant social, voluntary and charitable sector organisations; patient and public representatives as well as national and international research community representatives to discuss the results, further refine the framework for action, aid the write-up of the study findings and facilitate the translation of the findings into practice. The research will be disseminated via national and international academic and practitioner cross-over conferences, and a paper submitted to a leading journal in this field. Furthermore, a policy-orientated summary paper will be published on an open access basis so that it is freely available to practitioners and the public.

**Contributors** SS, CB, RC, RT and ML conceived this systematic review. SS, CB, RC, RT, ML, BN-M, JW and FB were involved in the planning and design of the study. SS registered the protocol in PROSPERO. BN-M and SS with support from FB undertook preliminary literature searches in MEDLINE. SS drafted and revised the manuscript. SS, CB, RC, RT, ML, BN-M, JW and FB have read, reviewed and contributed to revising and approving the final manuscript. SS obtained the research funding. SS is the guarantor.

**Funding** This work was supported by the National Institute for Health Research (NIHR) and Health Education England (HEE) Integrated Clinical Academic Lecturer Fellowship (ref CA-CL-2018-04-ST2-010) and Research Capability Funding, NHS North of England Commissioning Support (NECS). The systematic review is part of UNFAIR: UNderstanding Factors that explain Avoidable hospital admission Inequalities - Research study (http://bit.ly/UNFAIRstudy). This publication presents independent research funded by the National Institute for Health Research (NIHR) and Health Education England (HEE). The views expressed are those of the author(s) and not necessarily those of the NHS, the NIHR or the Department of Health and Social Care.

**Competing interests** None declared.

**Patient and public involvement** Patients and/or the public were involved in the design, or conduct, or reporting, or dissemination plans of this research. Refer to the Methods section for further details.

**Patient consent for publication** Not required.

**Provenance and peer review** Not commissioned; externally peer reviewed.

**ORCID iD**
Sarah Sowden http://orcid.org/0000-0001-9359-3463

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
