## [Reviewer comments · BMJ Open]

ARTICLE DETAILS

TITLE (PROVISIONAL)	Interventions to reduce inequalities in avoidable hospital admissions: explanatory framework and systematic review protocol
AUTHORS	Sowden, Sarah; Nezafat-Maldonado, Behrouz; Wildman, Josephine; Cookson, Richard; Thomson, Richard; Lambert, Mark; Beyer, Fiona; Bamba, Clare

VERSION 1 – REVIEW

REVIEWER	benedetta pongiglione Bocconi University, Centre for Research on Health and Social Care Management.
REVIEW RETURNED	24-Dec-2019

GENERAL COMMENTS	This systematic review protocol deals with an important topic: reviewing existing evidence of the effectiveness of interventions in reducing inequalities in avoidable admissions. Results of the review, either they will be produced as meta-analysis or narrative synthesis, will be very relevant to policy making and may have important implications. The rationale for the study, introduction and background are clearly stated. Methods to perform the systematic review illustrated in sufficient details. Hence, the work is suitable for publication conditional to some minor revisions. My main comment pertains to the outcome of interest. It is inconsistently and unclearly described throughout the text. In the title, objective (bottom page 3 of 13) and review questions (page 6 of 13) it is reported as avoidable (hospital) admissions. In the introduction it is stated that "(..) existing systematic reviews have examined only the effects of interventions on reducing levels of hospital admissions overall, as opposed to the effects on inequalities in avoidable hospital admissions" as further justification for this systematic review. However, in the methods section (Table 1 and "Outcomes" paragraph) there is no special focus on avoidable admission, and the search items for hospital admissions reported in Supplementary File 2 do not mention avoidable at all. This makes quite unclear what is the actual outcome of the review and how the search strategy is tailored to capture avoidable admission if this is the outcome of interest. Moreover in the whole text a definition of avoidable hospitalization is not provided, while literature has identified several (see for example Weissman, J.S., C. Gatsonis, and A.M. Epstein, Rates of avoidable hospitalization by insurance status in Massachusetts and Maryland. Jama, 1992. 268(17): p. 2388-2394; Billings, J. and R.M. Weinick, Monitoring the Health Care Safety Net: A Data Book for States and Counties. 2003) and this would also be a useful tool to compare outcomes from different research.
--

	Still related to the outcome, a minor comment relates the measures selected which are number or rates of hospital admission. Given the focus on comparing SES groups, some study may not (only) provide numbers and rates but also odds ratios, hazard rates or other measures and they should either be considered or a justification provided for solely focus on rates and numbers. The second main comment pertains to the fact that the review will include all studies on OECD countries, hence including countries with different healthcare systems and characteristics. The importance of context is remarked in the manuscript, for example when stated that “Context is increasingly recognised as important; real world interventions are rarely implemented in isolation, and the complex interaction of interventions with the particular social context in which they are embedded determines outcomes. (..) It is essential for policy and practice partners to learn not only that an intervention works to reduce inequalities in avoidable hospital per se, but why and how it managed to do this. (..) to compare the contextual situation to their own to make an assessment of both feasibility and likelihood of success of adopting a similar intervention in their specific socio-political, economic, cultural landscape.” My suggestion is to include a field in the data extraction specifically on the context and eventually to group the analysis by health care system (e.g. NHS, Bismarck, etc.) because a certain intervention, for example water fluoridation, can work very differently in a universal health care system compared to a insurance based. A few minor points: Figure 1: the box “Who stands to benefit?” has categories described in the text (pages 4 and 5 of 13) but unclearly labeled: what is the difference between lowest SE and lower SE? Study design/type described in Table 1 and “study design” paragraph (pages 6 and 7 of 13). Administrative data are an important source of data in this setting and may be worth explicitly mention it.
--	--

REVIEWER	Mirjam Allik SPHSU, UofG
REVIEW RETURNED	17-Jan-2020

GENERAL COMMENTS	The paper is overall clearly written and addresses a major public health concern, as well as could have strong financial and policy implications. Title – I wonder if it is possible to work in the word “interventions” to the title, so it would be explicit your work is on looking at interventions that attempt to reduce inequalities in admissions. Ethnicity – different ethnic groups may have different types of contact with the health services, and it may be that some (minority) populations, regardless of SES, might be more likely to have emergency and unplanned admissions. In some cases, SES and ethnicity may also overlap, so there may be confounding. Some attention/discussion of if/how diversity is included in these interventions is necessary. You may learn that one intervention worked for a specific ethnic group (or other sub-population) but will not work in another. You mention this very briefly in page 8, but I think it deserves more attention, given the diversity of most OECD nations. Maybe this is something that should be taken up in
---

	limitations also? Your studies may lack attention to important confounders that could have a big impact on the outcome. The relationship between socioeconomic inequalities in health and inequalities in avoidable admissions could use a bit more elaboration. Are you suggesting that reducing inequalities in avoidable admissions could lead to reduced health inequalities and this is why your work is important (page 3, policy context)? Or do reductions in health inequalities reduce inequalities in avoidable admissions (as per your figure)? Both can be feasible, but it would be nice to have these relationships made explicit. Study exclusion criteria – are you looking at countries that were members of OECD in 2019 or also in 1999 (some have joined between these dates)? Figure 1 – is the differentiation between lower and lowest important enough to separate these in box 2 (who benefits?). Or is this supposed to reflect the four different approaches outline on page 4 and 5 (if so, it is not clear).
--	--

REVIEWER	Beatriz Rodríguez-Sánchez Universidad de Castilla la Mancha, Spain
REVIEW RETURNED	22-Jan-2020

GENERAL COMMENTS	First of all, I would like to thank the authors and the editor for giving me the opportunity to read the systematic review protocol entitled "Reducing inequalities in avoidable hospital admissions: explanatory framework and systematic review protocol" submitted to BMJ Open for its potential publication. Although I have indeed enjoyed its reading, I do have some comments or concerns, which I hope might be of help for the authors. ABSTRACT  - From the title, it seems that the objective is two-fold: to develop an explanatory framework and to perform a systematic review. However, when I read the Methods and analysis section, it seems that the framework will be based on the results from the systematic review - Will economic evaluations, if any, be excluded from the review? - Are the authors going to compare middle-income countries vs high-income countries? The disease and the patient profiles might substantially differ between both populations and, hence, the comparisons should be interpreted with caution. This issue might motivate the possibility to "draw" a disadvantaged patient profile in the various settings. - Moreover, as the authors highlight in the limitations of the study, the definition of equity (and how it is measured) might change substantially between studies. INTRODUCTION  - Most references are focused on European Countries and, moreover, on the United Kingdom. Since OECD countries are not only European countries and socioeconomic disadvantages might be more present in other countries, these differences might be worthwhile to mention in the Introduction, comparing the results from published studies. Moreover, even in those countries, ethnicity could be a key driver of inequalities (Dalla Zuanna et al., 2017. Avoidable hospitalization among migrants and ethnic minority groups: a systematic review) - As stated before, why do the authors only focus on effectiveness and not on cost-effectiveness, for example?
---

	- Have the authors thought about a definition of socioeconomic groups, as socioeconomic group is mentioned in the four approaches? In order to foster comparability between studies METHODS AND ANALYSIS - It is not very clear to me how the inputs from PPIs and the findings from the systematic review will be merged - I would suggest the authors to include ethnicity or being an immigrant as one of the variables to include when referring to socioeconomic status. I can see that the authors aim to analyse the results by gender, age and ethnicity, but I would rather include ethnicity as one of the socioeconomic indicators, as education, income or deprivation are. - I would add another subgroup by grouping countries, according to, for example, GDP per capita or healthcare coverage. - I would add some additional information on the performance of the meta-analysis and how will equity be defined. It is not clear to me sometimes from the text, as one could interpret the systematic review as a review by socioeconomic groups, rather than in terms of equity differences.
--	---

REVIEWER	George Mnatzaganian La Trobe University, Australia
REVIEW RETURNED	23-Jan-2020

GENERAL COMMENTS	This proposed review aims to systematically review the literature to identify interventions that reduce socioeconomic inequalities in avoidable hospital admissions in OECD countries. - The aims of this SR are ambitious and not realistic. There are currently 36 countries that form the OECD with many having very different health systems (e.g., UK, USA, Turkey, Chile, Mexico, Israel, Greece, etc.). Some have national insurance coverage while others do not. To find interventions that will fit all is not feasible. - The way this paper is written is UK-centric. It would be a wiser step to make the objectives more specific and to only focus on the UK rather than all countries with so many differences. - The targeted population is way too broad and making this more specific would improve this review. Children are very different from the elderly. Medical admissions are very different from surgical ones. This review mixes all and does not seem to differentiate between these very different sub populations. - It is not clear what sort of interventions the authors are interested in? This proposed SR would significantly improve if the targeted interventions were made more specific. Socioeconomic disparities in health may be attributed to various determinants of health and health behaviours. Modifiable lifestyle and behavioural risk factors, such as poor diet, physical inactivity, smoking, and obesity often disproportionately affect individuals coming from the most disadvantaged groups. This proposed SR would significantly improve if say it would focus on interventions relating to modifiable risk factors such as smoking, BMI, exercise, diet, etc. - the study aims to assess the effectiveness of such interventions. Effectiveness is usually ascertained by clinical trials. However, this SR aims to include both clinical trials and observations studies. It
---

	is not clear how surveys or cross sectional data will answer this research question.  - The study outcome measures are not clear. The effect size is not clear. How will the differential effect be quantified? OR, RR, SMD? Or just pooled proportions using only studies that reported on SES disadvantaged groups? - Table 1 under “outcomes” has contradictory sentence under the exclusion. Since they authors aim to include studies that report only on one SES group, this exclusion criterion is incorrect. - Study design of “prospective repeat cross sectional” is an incorrect term. Longitudinal repeat cross sectional studies are called “panel”. Please delete “prospective repeat cross sectional”. - Why studies reporting time series data will not be considered in this very broad study? - A minor comment relates to timelines listed under methods and analysis (lines 36,37). We are already approaching February 2020. To say that “the review is planned to commence November 2019” is inaccurate. Please correct.
--	--

VERSION 1 – AUTHOR RESPONSE

Reviewer: 1 Benedetta Pongiglione, Bocconi University, Centre for Research on Health and Social Care Management.

2. “This systematic review protocol deals with an important topic: reviewing existing evidence of the effectiveness of interventions in reducing inequalities in avoidable admissions. Results of the review, either they will be produced as meta-analysis or narrative synthesis, will be very relevant to policy making and may have important implications. The rationale for the study, introduction and background are clearly stated. Methods to perform the systematic review illustrated in sufficient details. Hence, the work is suitable for publication conditional to some minor revisions.”

Thank you for this assessment. We have made revisions in response to your feedback as follows.

3. “My main comment pertains to the outcome of interest. It is inconsistently and unclearly described throughout the text....Moreover in the whole text a definition of avoidable hospitalization is not provided.”

Thank you for this point and for the suggested references, we have now included Billings et al. reference in the manuscript. We have addressed this comment by a) improving the consistency of terminology used throughout the manuscript b) providing a clarification of the definition of avoidable hospital admissions we will use in this review (lines 271-284) and c) have provided references to a range of relevant definitions and technical specifications to demonstrate the permissive nature of the definition we will adopt in this review to ensure important empirical data from different counties are captured.

4. “Still related to the outcome, a minor comment relates the measures selected which are number or rates of hospital admission. Given the focus on comparing SES groups, some study may not (only) provide numbers and rates but also odds ratios, hazard rates or other measures and they should either be considered or a justification provided for solely focus on rates and numbers.”

Thank you, we are including all appropriate measures and have expanded the range of outcomes listed to reflect this (lines 289-292) and in table 1. We have also ensured the data extraction form for the review includes the array of potential ways in which inequality in outcomes may be measured. We

have included a copy of the data extraction form as a supplementary file with the resubmission.

5. “The second main comment pertains to the fact that the review will include all studies on OECD countries, hence including countries with different healthcare systems and characteristics.... My suggestion is to include a field in the data extraction specifically on the context and eventually to group the analysis by health care system (e.g. NHS, Bismarck, etc.) because a certain intervention, for example water fluoridation, can work very differently in a universal health care system compared to a insurance based.”

Thank you for raising this important point. We have added reference to this in the manuscript: abstract (lines 55-56), methods (lines 297-299) “We recognise that OECD countries will have different characteristics including different health systems and socio-economic situations and we will interpret the interventions that we find taking into account this variation in context.” We have included ‘health care system’ as a specific data collection field within the data collection form and have paid close attention to ensuring we have data collection fields to capture a range of relevant contextual information to assist in the interpretation of findings (see data collection form in supplementary file 3) and explanation in the manuscript (lines 337, 349-353).

6. “Figure 1: the box “Who stands to benefit?” has categories described in the text (pages 4 and 5 of 13) but unclearly labeled: what is the difference between lowest SE and lower SE?”

Figure 1 has been revised so that the wording directly corresponds to the wording used in lines 187-198 of the manuscript to which it directly relates. We have added the sub-headings ‘domain of action’ and ‘approach’ to more clearly demonstrate the direct relationship between the framework diagram and the manuscript (lines 189-198). We have changed the wording not use ‘lowest’ and ‘lower’ SE to make things clearer and also to standardise terminology used throughout the manuscript when referring to those from different socioeconomic groups.

7. “Study design/type described in Table 1 and “study design” paragraph (pages 6 and 7 of 13).

Administrative data are an important source of data in this setting and may be worth explicitly mention it.”

Thank you, we have added an explicit mention of studies using administrative data (line 306).

Reviewer 2 - Mirjam Allik, Social and Public Health Sciences Unit, University of Glasgow

8. “The paper is overall clearly written and addresses a major public health concern, as well as could have strong financial and policy implications.”

Thank you for this assessment. We have made revision in response to your feedback as follows.

9. “Title – I wonder if it is possible to work in the word “interventions” to the title, so it would be explicit your work is on looking at interventions that attempt to reduce inequalities in admissions.”

Thank you, we have changed the title to: “Interventions to reduce inequalities in avoidable hospital admissions: explanatory framework and systematic review protocol” (lines 1-2)

10. “Ethnicity - different ethnic groups may have different types of contact with the health services, and it may be that some (minority) populations, regardless of SES, might be more likely to have emergency and unplanned admissions. In some cases, SES and ethnicity may also overlap, so there may be confounding. Some attention/discussion of if/how diversity is included in these interventions is necessary. You may learn that one intervention worked for a specific ethnic group (or other sub-population) but will not work in another. You mention this very briefly in page 8, but I think it deserves more attention, given the diversity of most OECD nations. Maybe this is something that should be taken up in limitations also? Your studies may lack attention to important confounders that could have a big impact on the outcome.”

We absolutely agree here. As the reviewer notes there will be a relationship between ethnicity and socioeconomic status – it may serve as an effect modifier or confounder, and the relationship is likely

to be context specific and may vary across country/health care system etc. As suggested we have given more attention to population characteristics (specifically ethnicity) in the manuscript as follows: included in the abstract (lines 55-56), introduction (lines 91-92, added useful reference here suggested by reviewer 3, thank you), methods (lines 338-339), data collection form (see supplementary file 3 included with the resubmission), and the limitations section of the discussion (lines 412-417).

11. "The relationship between socioeconomic inequalities in health and inequalities in avoidable admissions could use a bit more elaboration. Are you suggesting that reducing inequalities in avoidable admissions could lead to reduced health inequalities and this is why your work is important (page 3, policy context)? Or do reductions in health inequalities reduce inequalities in avoidable admissions (as per your figure)? Both can be feasible, but it would be nice to have these relationships made explicit."

It is the latter we have focused on principally (reductions in health inequalities will lead to reductions in inequalities in avoidable admissions). We were making a similar point in the policy context section (to reduce hospital costs you need to reduce health inequalities), alongside the allied goal of reducing inequalities in healthcare access, use and outcome (e.g. the Spanish and English legislative examples). "And healthcare" has been added (line 108) to make this clearer. Lines 149-151 explain that the framework is a starting point and will undergo further iterative development based on the findings of the review and also based on input from patient and professional experts. The causal feedback loops existing in this complex system will be an area for further exploration as the work progresses.

12. "Study exclusion criteria – are you looking at countries that were members of OECD in 2019 or also in 1999 (some have joined between these dates)?"

We are including studies conducted in any country that is an OECD member as of 28 Feb 2020 (the revised end date for the search). As countries become candidates before members of the OECD and follow an accession process we believe that studies published before joining the OECD may be of interest for our study. We recognise that OECD countries will vary markedly and have ensured the manuscript, data collection process and data analysis will take this in to account (please see changes made in our response to point 5 by reviewer 1 above).

13. "Figure 1 – is the differentiation between lower and lowest important enough to separate these in box 2 (who benefits?). Or is this supposed to reflect the four different approaches outline on page 4 and 5 (if so, it is not clear)."

Thank you, reviewer 1 commented on this too (please see changed made in our response to point 6 by reviewer 1 above).

Reviewer: 3 - Beatriz Rodríguez-Sánchez, Universidad de Castilla la Mancha, Spain

14. "From the title, it seems that the objective is two-fold: to develop an explanatory framework and to perform a systematic review. However, when I read the Methods and analysis section, it seems that the framework will be based on the results from the systematic review."

In line with PRISMA-E guidelines, the first stage for any equity-focused review is to develop a logic model/framework at the outset and so in accordance with this guideline a framework has been included in the manuscript (lines 143-149). However, this is a starting point. The intention is for this framework to be further revised and improved iteratively as evidence from the systematic review emerges and in consultation with patient, public, and policy and practice partners (lines 149-154).

15. "Will economic evaluations, if any, be excluded from the review?"

Economic evaluations will be included if they are within study type inclusion (i.e. experimental, quasi-

experimental or a causal-inference type study) and have measured the outcome variable of interest. In this review we are not looking at secondary outcome measures such as cost/cost effectiveness of interventions.

16. "Are the authors going to compare middle-income countries vs high-income countries? The disease and the patient profiles might substantially differ between both populations and, hence, the comparisons should be interpreted with caution."

Thank you, reviewer 1 makes a similar point. Please see changes made in our response to point 5 by reviewer 1 above.

17. "Moreover, as the authors highlight in the limitations of the study, the definition of equity (and how it is measured) might change substantially between studies."

Yes, we have included an additional reference to this point in the data extraction section (lines 343-345).

18. "Most references are focused on European Countries and, moreover, on the United Kingdom. Since OECD countries are not only European countries and socioeconomic disadvantages might be more present in other countries, these differences might be worthwhile to mention in the Introduction, comparing the results from published studies."

Thank you – reviewer 1 makes a related point. Please see changes made in our response to point 5 by reviewer 1 above.

19. "Moreover, even in those countries, ethnicity could be a key driver of inequalities (Dalla Zuanna et al., 2017. Avoidable hospitalization among migrants and ethnic minority groups: a systematic review)"

Thank you, we have included this reference. Please see changes made in response to point 10, reviewer 2, outlined above.

20. "As stated before, why do the authors only focus on effectiveness and not on cost-effectiveness, for example?"

We did not feel it is feasible in the capacity constraints for this review to extend the scope to include a consideration of secondary research questions relating to, for example, cost effectiveness. That said, as explained in response to point 15, economic evaluations will be included within the scope of the review if they are an included study design and present data for the outcome variable of interest for this review.

21. "Have the authors thought about a definition of socioeconomic groups, as socioeconomic group is mentioned in the four approaches?"

We expect authors of primary studies to use a range of definitions and methods for characterising socioeconomic groups (added mention of this lines 343-345). Therefore, we have kept our definition (in terms of the ways in which socioeconomic status may be characterised by studies) deliberately inclusive and broad (lines 286-288). Through the data collection tool (see supplementary file 3 included in this resubmission) we aim to record the findings from studies clearly and precisely in terms of how they are measuring socioeconomic status and intend to use this contextual information when interpreting the findings of the review.

22. "It is not very clear to me how the inputs from PPIs and the findings from the systematic review will be merged."

The intention is that this merging activity will occur through discussion at the collaborator advisory panel meetings where input from experts (patients and professionals alike) in conjunction with the evidence that is emerging from the review will be considered (lines 239-242).

23. "I would suggest the authors to include ethnicity or being an immigrant as one of the variables to

include when referring to socioeconomic status. I can see that the authors aim to analyse the results by gender, age and ethnicity, but I would rather include ethnicity as one of the socioeconomic indicators, as education, income or deprivation.”

Please see response to point 10 (reviewer 1) and point 19 outlining the modifications we have made to the manuscript in light of reviewer feedback around ethnicity.

24. “I would add another subgroup by grouping countries, according to, for example, GDP per capita or healthcare coverage.”

Thank you, please see response to point 5 (reviewer 1) point 16 and point 18 which address this.

25. “I would add some additional information on the performance of the meta-analysis and how will equity be defined. It is not clear to me sometimes from the text, as one could interpret the systematic review as a review by socioeconomic groups, rather than in terms of equity differences.”

The review will be looking at outcome of either a targeted intervention, or a universal intervention where differential effectiveness of the intervention across socioeconomic group has been considered. If studies are sufficiently similar in PICO (assessed on basis of detailed data collection proforma, see supplementary file 3). Studies may be combined in meta-analysis. Based on this analysis, an assessment of the equity or otherwise of the performance of given interventions will be made in line with the classification outlined in the framework (lines 199-217 provides examples).

Reviewer: 4 - George Mnatzaganian, La Trobe University, Australia

26. “The aims of this SR are ambitious and not realistic. There are currently 36 countries that form the OECD with many having very different health systems (e.g., UK, USA, Turkey, Chile, Mexico, Israel, Greece, etc.). Some have national insurance coverage while others do not. To find interventions that will fit all is not feasible.”

In response to this and related comments we have enhanced the prominence and discussion about how variability in context will be handled in the review – please see changed made to the manuscript in our response to point 5 (reviewer 1), point 16, point 18 and point 24 (reviewer 3) which address this and related points.

27. “The way this paper is written is UK-centric. It would be a wiser step to make the objectives more specific and to only focus on the UK rather than all countries with so many differences.”

Thank you, we have considered the suggestion to restrict to UK only but have decided to keep the international focus for two reasons. Firstly, preliminary searches (lines 223-226) indicated a limited number of papers which have either looked at differential effectiveness of interventions across socioeconomic groupings, or have been targeted specifically at disadvantaged groups per se. Therefore a review restricted to only UK studies is unlikely to generate a sufficient body of literature to review. Secondly, and more fundamentally, whilst acknowledging the importance of context, and the need for careful consideration of this when assessing the comparability and generalisability of findings, the UK can learn from interventions implemented in other countries and vice versa. Therefore, we believe maintaining the international focus for the review is important. Within the introduction, in addition references to Spain (lines 108-112), OECD countries (lines 101, lines 104) and WHO (104-107), we have added lines 89-92 to further strengthen the justification for this international focus. We have re-phrased throughout (for example in discussion of definition of avoidable hospital admission, lines 271-277) in an attempt to use international terminology suitable for an international readership.

28. “The targeted population is way too broad and making this more specific would improve this review. Children are very different from the elderly. Medical admissions are very different from surgical ones. This review mixes all and does not seem to differentiate between these very different sub populations.”

The rationale for including a broad study population is so that it corresponds to the populations for the outcome of interest (hospitalisations for ambulatory care sensitive conditions). This necessarily includes patients with a range of clinical conditions across ages (children and adults). In practice, given the majority of hospitalisations are for the older adult population, our findings will be predominantly covering the older adult population. Nevertheless, we will attempt to report results separately for different child and elderly populations where available and pay attention to age as an explanatory factor (lines 389-395). The inclusion of a broad population is also required to ensure alignment to the systems perspective for the review (lines 128-134). The lack of this whole systems integrative perspective and contextual considerations was cited as a limitation of previous avoidable hospital admission literature (see manuscript ref 12).

29. "It is not clear what sort of interventions the authors are interested in? This proposed SR would significantly improve if the targeted interventions were made more specific."

The sort of interventions the review is interested in are categorised under four domains of action (1-4, see left-hand side of figure 1), are explained in the introduction (lines 153-186) and methods (lines 262-269) sections of the review, with examples of specific interventions provided (lines 116-117, lines 171-186).

30. "Socioeconomic disparities in health may be attributed to various determinants of health and health behaviours. Modifiable lifestyle and behavioural risk factors, such as poor diet, physical inactivity, smoking, and obesity often disproportionately affect individuals coming from the most disadvantaged groups. This proposed SR would significantly improve if say it would focus on interventions relating to modifiable risk factors such as smoking, BMI, exercise, diet, etc."

Thank you and yes we will pay close attention to these kinds of interventions targeting health behaviours (which depending on the level and nature of operation will fall under any one of the 4 domains of activity) as these will be relevant to the top right hand side of the framework (figure 1). Please see also response to point 29 above. We aim however to look broader than this; the intention is for the review to adopt a systems level perspective and look at action across all domains of activity relevant to the framework for action. Therefore any literature on other interventions which operate in the bottom right hand side of the framework (figure 1) i.e. interventions to address the differential propensity of more disadvantaged socioeconomic groups to use planned as opposed to emergency healthcare will be included.

31. "The study aims to assess the effectiveness of such interventions. Effectiveness is usually ascertained by clinical trials. However, this SR aims to include both clinical trials and observations studies. It is not clear how surveys or cross sectional data will answer this research question."

The rationale for inclusion of studies wider than clinical trials to encompass quasi-experimental and causal inference studies is provided in the manuscript (lines 418-422) and additional information on how these non-experimental studies will be handled has now been provided in lines 367-371. In response to the reviewer comment, we have also included an additional reference that supports the justification of inclusion of studies using non-experimental designs (if available) when considering interventions across all four domains of action (Ogilvie et al, JECH 2019, doi: 10.1136/jech-2019-213085). We agree with the reviewer that purely descriptive cross-sectional studies will not answer the research questions posed in this review and have indicated that these studies will be excluded from the review (Table 1 – exclusion: descriptive studies reporting solely on avoidable admission prevalence (by person, place and time)).

32. "The study outcome measures are not clear. The effect size is not clear. How will the differential effect be quantified? OR, RR, SMD? Or just pooled proportions using only studies that reported on SES disadvantaged groups?"

Thank you, reviewer 1 makes a similar point, please see changed made in our response to point 4 by reviewer 1 above.

33. "Table 1 under "outcomes" has contradictory sentence under the exclusion. Since they authors aim to include studies that report only on one SES group, this exclusion criterion is incorrect." Thank you for alerting us to this – the wording has been amended to make it clear that this criteria is N/A for studies of interventions targeted exclusively at socioeconomically disadvantaged populations.

34. "Study design of "prospective repeat cross sectional" is an incorrect term. Longitudinal repeat cross sectional studies are called "panel". Please delete "prospective repeat cross sectional". Why studies reporting time series data will not be considered in this very broad study?" Thank you, wording of manuscript has been amended to ensure these types of studies are included and correctly labelled (lines 301-307 and table 1).

35. "A minor comment relates to timelines listed under methods and analysis (lines 36,37). We are already approaching February 2020. To say that "the review is planned to commence November 2019" is inaccurate. Please correct."

We have amended the start date to March 2020 (line 225) and have revised the 20 year search timeframe accordingly (search to include studies published 2000 onwards – line 294 and table 1).

General

36. "Corresponding author email address in ScholarOne system is different from the main document. Kindly amend accordingly."

Thank you, we have amended the corresponding author email address on manuscript so that it matches the ScholarOne system information. We have also updated the Institute name for authors from one university as tis has recently changed.

VERSION 2 – REVIEW

REVIEWER	benedetta pongiglione Bocconi University, Centre for Research on Health and Social Care Management
REVIEW RETURNED	08-Mar-2020

GENERAL COMMENTS	Thank you to the authors for responding constructively to most of the comments. Overall, replies are satisfactory, but there remains a couple of points that could be addressed to improve the manuscript. In the attempt of addressing the reviewers' comments, some parts of the text have become convoluted and the reading is not always smooth. This applies to the abstract, in particular the second paragraph (from line 38). Also the first section of the introduction needs some re-writing. In line 89, you introduce quite abruptly OECD countries, before you illustrate that you will focus on such countries, and in the same paragraph, without a clear flow, you mention some specific socioeconomic inequalities. The other comment pertains to the types of studies that you intend to include. In the "study design" paragraph you now write "A rigorous and inclusive international literature search for experimental, quasi-experimental and causal inference based studies will be conducted". There seems to be some confusion between study design of interest and methodologies. My understanding is that you want to assess if findings from non-RCTs studies can be considered causal, hence including such type of study and assessing their quality and risk of bias. If so, please reframe the paragraph study design. Otherwise, if you intend to include only non-RCTs that use causal inference
--

	techniques, you should include this among the inclusion criteria and more details should be given on how you will apply it. Please be clearer about it and consider possible implications. Minor remarks: Abstract: unclear to what refers “for whom” at line 54 Abstract: avoid using abbreviations such as “doesn’t” Please provide a reference for the statement from line 162 and 166 Table 1: please present the list of inclusion criteria for outcomes more clearly, either using bullet point or improving text Line 296: the acronym for OECD compares fully spelt here for the first time I do not think you need the sentence “The skills of a trained information scientist (FB) will be used to implement the electronic searches” The sentence you added at lines 343-345 has repetitions and does not read well. Supplement 3: can you please provide a reference for the OECD survey of health system characteristics. The types of coverage are not those commonly used.
--	---

REVIEWER	Mirjam Allik SPHSU, UofG
REVIEW RETURNED	11-Mar-2020

GENERAL COMMENTS	The authors have given due considerations to the comments I made and have revised their manuscript accordingly. Good luck with the research.
--

REVIEWER	Beatriz Rodríguez Sánchez University of Castilla la Mancha, Spain
REVIEW RETURNED	16-Mar-2020

GENERAL COMMENTS	I would like to congratulate the authors for the improvement made to the manuscript. All my comments and suggestions have been addressed.
---

REVIEWER	George Mnatzaganian La Trobe University, Australia
REVIEW RETURNED	08-Mar-2020

GENERAL COMMENTS	1. I thank the reviewers for the revision. My first comment was not addressed; I will refer to it my second point. I appreciate the authors to re-write what they stated for previous reviewers. It is not reader friendly to search for answers in “point 5 (reviewer 1), point 16, point 18 and point 24 (reviewer 3)”. It is preferred if the authors could respond and re-write their responses to each of the comments for each reviewer. I am reviewer 4. You may end up asking me to always look back and see what you have written previously. Thank you. 2. These responses did not address my comments properly. The aims of this SR are still ambitious and not realistic. As stated earlier, OECD economies have different health systems and the study main outcome, “avoidable admissions”, may be defined differently by each OECD country.
--

	If the authors wish to proceed with this broad inclusion of all countries, I advise them to include an appendix in which they will list all 36 OECD economies together with some basic characteristics of each country's health system including national health coverage and GDP per capita and health consumption spending per capita, and a few indicators related to health resources such as number of hospital beds and number of GPs and nurses per population. Such an appendix can shed light on the similarities and differences among the different countries represented. The chart presented in https://data.oecd.org/healthres/health-spending.htm demonstrates the point I am making. This project aims to compare apples with oranges under the naïve assumption that OECD health systems are comparable. And it is not about the "UK learning from interventions implemented in other countries and vice versa", as many of the health systems in the OECD countries are very different and non-comparable. I again advise to make the focus of this review less UK-centric. It was and continues to be UK-centric. 3. This proposed SR is spread too thin, being too broad, too general and ambitiously unrealistic aiming to compare the incomparable. It is not the issue of the "UK learning from other countries and vice versa" but rather the UK, Turkey, India, Chile, Brazil, Russia and all other OECD economies are enormously different. Because of this inherent heterogeneity, I doubt it if the authors will be able to produce anything except a narrative synthesis.
--	--

VERSION 2 – AUTHOR RESPONSE

Reviewer: 1 Benedetta Pongiglione, Bocconi University, Centre for Research on Health and Social Care Management.	
Thank you to the authors for responding constructively to most of the comments. Overall, replies are satisfactory, but there remains a couple of points that could be addressed to improve the manuscript.	Thank you.
In the attempt of addressing the reviewers' comments, some parts of the text have become convoluted and the reading is not always smooth. This applies to the abstract, in particular the second paragraph (from line 38).	Thank you. To make the text in the abstract less convoluted we have added an extra full stop to break up the text and reordered the words (lines 42-43).
Also the first section of the introduction needs some re-writing. In line 89, you introduce quite abruptly OECD countries, before you illustrate that you will focus on such countries, and in the same paragraph, without a clear flow, you mention some	Thank you for this advice. We have rewritten lines 81-92 in order to improve the flow and take out the abrupt first reference to OECD countries.

specific socioeconomic inequalities.	
The other comment pertains to the types of studies that you intend to include. In the “study design” paragraph you now write “A rigorous and inclusive international literature search for experimental, quasi-experimental and causal inference based studies will be conducted”. There seems to be some confusion between study design of interest and methodologies. My understanding is that you want to assess if findings from non-RCTs studies can be considered causal, hence including such type of study and assessing their quality and risk of bias. If so, please reframe the paragraph study design.	“My understanding is that you want to assess if findings from non-RCTs studies can be considered causal, hence including such type of study and assessing their quality and risk of bias.” Yes. This is correct. If so, please reframe the paragraph study design. Thank you. We have now added lines 301-303 to the study design paragraph to reframe this: “In the case of non-RCT studies the authors will make a judgement as to whether any association reported may be considered causal, taking into account the type of study undertaken and following a thorough assessment of study quality and risk of bias (see risk of bias section below).”
Abstract: unclear to what refers “for whom” at line 54	“for whom” refers to which individuals/population the intervention has been shown to be effective for. For example, a given intervention may be effective at reducing socioeconomic inequalities in hospital admissions in children with asthma, or males over the aged of 65 with angina.
Abstract: avoid using abbreviations such as “doesn’t”	Thank you. This has been changed to ‘does not’.
Please provide a reference for the statement from line 162 and 166	Reference has been added.
Table 1: please present the list of inclusion criteria for outcomes more clearly, either using bullet point or improving text	Clarity has been improved by including a bullet point list. We have also modified other cells in table 1 to include bulleted lists to ensure consistency in presentation across the table.
Line 296: the acronym for OECD compares fully spelt here for the first time	Thank you. We have ensured the acronym for OECD is spelt out on the first occasion it is mentioned in the manuscript.
I do not think you need the sentence “The skills of a trained information scientist (FB) will be used to implement the electronic searches”	This has been removed.
The sentence you added at lines 343-345 has repetitions and does not read well.	Thank you. We have made this sentence read better by breaking it up into two sentences and rewording to remove repetition.
Supplement 3: can you please provide a reference for the OECD survey of health system characteristics.	This has been added in Supplement 3, thank you.
Reviewer 2 - Mirjam Allik, Social and Public Health Sciences Unit, University of Glasgow	

The authors have given due considerations to the comments I made and have revised their manuscript accordingly. Good luck with the research.	Thank you.
Reviewer: 3 - Beatriz Rodríguez-Sánchez, Universidad de Castilla la Mancha, Spain	
I would like to congratulate the authors for the improvement made to the manuscript. All my comments and suggestions have been addressed.	Thank you.
Reviewer: 4 - George Mnatzaganian, La Trobe University, Australia	
The aims of this SR are still ambitious and not realistic.	We agree the aims of this SR are ambitious. However, we believe it is realistic. Pilot search strategy testing has indicated there is a body of literature to appraise which is not too overwhelmingly large and diverse to review, and previous research has indicated a gap in the knowledge base in this area, including the need for both an equity perspective and a systems level approach looking at interventions across domains of action. We have taken many steps to develop and refine the PICO for this systematic review to ensure we are conducting a realistic piece of research, guided in part by the insightful feedback from reviewer 4 and the other reviewers and we would like to thank them for their assistance in helping us to improve the manuscript, and in turn the planned research.
As stated earlier, OECD economies have different health systems and the study main outcome, “avoidable admissions”, may be defined differently by each OECD country.	Thank you, we agree with the reviewer on both points. OECD economies have different health systems and this context needs careful attention when interpreting findings – especially but not only the large system-level differences between middle income countries like Turkey versus high-income countries like the UK. We have made reference to this in the manuscript. For example: Abstract lines 54-55; “contextual information on where...interventions are organised, implemented and delivered will be examined” Introduction lines 132-133; “the complex interaction of interventions with the particular context in which they are embedded determines outcomes” Methods lines 292-294; “we recognise that OECD countries will have different characteristics including different health systems and socio-economic situations and we will interpret the interventions that we find taking into account this variation in context”. We intend to capture this important contextual information in our data collection (see supplementary file 3). We have further enhanced our consideration of this context by including a recently developed typology for classifying OECD healthcare systems within the data collection tool (reference 56 https://www.sciencedirect.com/science/article/pii/S0168851019301083). We also agree with the reviewer that the main outcome ‘avoidable admissions’ may be defined differently by each OECD country, and that this also needs careful attention when

	interpreting findings. We have made reference to this in the manuscript: ‘Outcomes’ section lines 273-278; “Different studies use different lists of ambulatory care sensitive conditions, reflecting different types of illness...different types of care...and differences in clinical coding definitions. Avoidable admissions are sometimes referred to as preventable admissions. Our definition of outcome is general and permissive of all this definitional variation within and between countries”.
If the authors wish to proceed with this broad inclusion of all countries, I advise them to include an appendix in which they will list all 36 OECD economies together with some basic characteristics of each country’s health system including national health coverage and GDP per capita and health consumption spending per capita, and a few indicators related to health resources such as number of hospital beds and number of GPs and nurses per population. Such an appendix can shed light on the similarities and differences among the different countries represented. The chart presented in https://data.oecd.org/healthres/health-spending.htm demonstrates the point I am making.	Thank you for this suggestion. We have not included our own bespoke health system country summary in an appendix as suggested by reviewer 4. Instead, to directly address this point we have provided reference (reference 57 – OECD Health System Characteristics survey) to a comprehensive survey of all OECD countries which includes information on the characteristic’s reviewer 4 rightly highlights as important. The reference supplied also includes further onward referencing to other documentation providing detailed information on other OECD characteristics. In addition, we have now included another reference to a recently developed typology for classifying OECD country healthcare systems, which will be used within the data collection tool (reference 56 https://www.sciencedirect.com/science/article/pii/S0168851019301083). We agree that when it comes to writing up the review itself (as opposed to the protocol), it may be useful to include bespoke summaries of key health system context to aid interpretation of our findings.
This project aims to compare apples with oranges under the naïve assumption that OECD health systems are comparable. And it is not about the “UK learning from interventions implemented in other countries and vice versa”, as many of the health systems in the OECD countries are very different and non-comparable.	We fully appreciate and agree with the reviewer that it will be challenging to generalise given the context specific nature of many of the interventions we will be considering. This issue is not unique to this proposed systematic review. We have however, been very clear in the manuscript that context will be carefully considered and taken in to account when analysing the results and reporting the findings. For example: Abstract lines 54-55; “contextual information on where...interventions are organised, implemented and delivered will be examined” Introduction lines 132-133; “the complex interaction of interventions with the particular context in which they are embedded determines outcomes” Methods lines 292-294; “we recognise that OECD countries will have difference characteristics including different health systems and socio-economic situations and we will interpret the interventions that we find taking into account this variation in context”. We intend to capture this important contextual information in our data collection (see supplementary file 3) and will if feasible carry out a synthesis by health system type using a recently developed typology for classifying OECD health systems (Reference 56 https://www.sciencedirect.com/science/article/pii/S0168851019301083).

I again advise to make the focus of this review less UK-centric. It was and continues to be UK-centric.	We apologise if our language appears UK-centric, and have done our best to internationalise the language – for example, we include background information from countries other than the UK (e.g. Spain) and, on the back of previous very helpful comments from reviewer 4 and others, we have taken further steps to ensure the international focus and relevance of this systematic review is presented in the manuscript. Our intention is to make this an internationally useful review. By including studies conducted in all OECD countries in the search strategy we are making this systematic review as international as possible, with the necessary caveats outlined in point 16 above around ensuring context is very carefully considered and taken in to account when analysing the results and reporting the findings.
This proposed SR is spread too thin, being too broad, too general and ambitiously unrealistic aiming to compare the incomparable. It is not the issue of the "UK learning from other countries and vice versa" but rather the UK, Turkey, India, Chile, Brazil, Russia and all other OECD economies are enormously different. Because of this inherent heterogeneity, I doubt it if the authors will be able to produce anything except a narrative synthesis.	This comment summarises comments made in points 12 – 16 above which we have addressed in turn above. We agree with the reviewer that it may be the case that we will only be able to produce a narrative synthesis of the literature and this is explained in the manuscript (lines 50-51 abstract: "Narrative synthesis will be conducted and also meta-analysis where possible", lines 378-380 data analysis: "Data will be summarised statistically using meta-analysis of effect measures across socioeconomic sub-groups, where appropriate, and if studies are deemed sufficiently homogeneous to combine.").

VERSION 3 – REVIEW

REVIEWER	Benedetta Pongiglione Centre for Research on Health and Social Care Management, SDA Bocconi.
REVIEW RETURNED	07-Apr-2020
GENERAL COMMENTS	Thank you to the authors for addressing all the minor comments. Replies are satisfactory, and the paper is suitable for publication. The only suggestion I have is to change the term "causal inference based studies" with another term which describes the study design and not the methodology, for example using "observational studies" or "non-RCT studies" or "non interventional studies" in the sentence reported in line 295-296 "A rigorous and inclusive international literature search for experimental, quasi-experimental and causal inference based studies will be conducted. "